# Pain and Analgesia in Children with Cancer after Hemipelvectomy: A Retrospective Analysis

**DOI:** 10.3390/children9020237

**Published:** 2022-02-11

**Authors:** Vamshi R. Revuri, Karen Moody, Valerae Lewis, Rodrigo Mejia, Douglas J. Harrison, Ali H. Ahmad

**Affiliations:** 1Pediatric Critical Care Fellowship Program, Department of Pediatric Critical Care Medicine, The University of Texas Health Science Center at Houston, Children’s Memorial Hermann Hospital, Houston, TX 77030, USA; vamshi.raman.revuri@uth.tmc.edu; 2Pediatric Palliative and Supportive Care, Department of Pediatrics, The University of Texas MD Anderson Cancer Center, Houston, TX 77030, USA; kmoody@mdanderson.org; 3Department of Orthopaedic Oncology, The University of Texas MD Anderson Cancer Center, Houston, TX 77030, USA; volewis@mdanderson.org; 4Section of Pediatric Critical Care, Department of Pediatrics, The University of Texas MD Anderson Cancer Center, Houston, TX 77030, USA; rmejia@mdanderson.org; 5Department of Pediatrics-Patient Care, The University of Texas MD Anderson Cancer Center, Houston, TX 77030, USA; djharrison@mdanderson.org

**Keywords:** pediatrics, cancer, pain, analgesia, oncology, orthopedics, critical care

## Abstract

A paucity of data exists centering on the pain experience of children following hemipelvectomy performed for primary bone and soft tissue sarcomas. In this study, we aimed to describe the incidence, severity, and evolution of perioperative pain and function in pediatric oncology patients undergoing hemipelvectomy, and, additionally, we sought to detail the analgesic regimens used for these patients perioperatively. A retrospective chart review was conducted, studying cancer patients, aged 21 years and under, who underwent hemipelvectomy at MD Anderson Cancer Center (MDACC) from 2018 to 2021. Primary outcomes included the evolution of pain throughout the perioperative course, as well as the route, type, dose, and duration of analgesic regimens. Eight patients were included in the analysis. The mean age at operation was 13 ± 2.93 years. All patients received opioids and acetaminophen. The mean pain scores were highest on post-operative day (POD)0, POD5, and POD 30. The mean opioid use was highest on POD5. A total of 75% of patients were noted to be ambulating after hemipelvectomy. The mean time to ambulation was 5.33 ± 2.94 days. The combination of acetaminophen with opioids, as well as adjunctive regional analgesia, non-steroidal anti-inflammatory drugs, gabapentin, and/or ketamine in select patients, appeared to be an effective analgesic regimen, and functional outcomes were excellent in 75% of patients.

## 1. Introduction

Hemipelvectomy is a complex surgical procedure performed for malignant or locally aggressive pelvic bone and soft tissue tumors, and it is less commonly performed for trauma or osteomyelitis [1,2,3,4]. Regarding oncologic diagnoses, hemipelvectomy is performed by highly specialized orthopedic oncology teams at tertiary or quaternary institutions [5]. Osteosarcoma and Ewing’s sarcoma are the most common malignant bone tumors of the pelvis in children that present for hemipelvectomy [6,7,8,9]. Treatment for these malignant neoplasms is multifactorial, including both systemic chemotherapy and surgery, as well as radiation therapy, depending on the specific histologic diagnosis [6]. The most common presenting symptoms of a pelvic tumor are related to mass effect from the mass itself, and these symptoms include pain, deep venous thrombosis, bladder or bowel dysfunction, and constitutional symptoms, such as fever or weight loss [6,8,9]. Sarcomas within the pelvis are associated with a worse prognosis than primary tumors of the extremity of the same histology [10]. As pelvic tumors grow, they often encroach upon the other organs within the pelvis, such as the bowel, bladder, vagina, prostate, and/or neurovascular bundles [1,2,8]. It is not entirely clear whether the worse prognosis is due to the intrinsic biology of these tumors, or to the fact that negative margins are more difficult to obtain due to the location. Pelvic tumors require aggressive local control, involving internal or external hemipelvectomy with resection of the encroaching or compromised organs, in order to optimize survival [11]. In addition, in select cases, both internal and external hemipelvectomy can offer a palliative approach to the management of pain, pathological fractures, infection, and fungating wounds [5].

Although external hemipelvectomy (EH) was once the surgical approach of choice for the treatment of pelvic sarcomas, limb-sparing internal hemipelvectomy (IH), when possible, has now become the popular and more viable option [7,10]. There are three important structures in the pelvis that are necessary in order to maintain a functional limb: the lumbar-sacral plexus, the acetabulum, and the femoral neurovascular bundle [10,11,12,13]. Should the tumor compromise and/or invade two of these structures, then the limb should not be salvaged, as it will not be functional [1,2]. However, if the neoplasm only compromises one of these structures, an internal hemipelvectomy (IH) may be performed [1,2]. In 1978, Enneking et al. established a classification system for pelvic resections based on the area of the pelvis resected: Type 1 involves the ilium; Type 2 involves the acetabulum/periacetabulum; Type 3 involves the pubis; and Type 4 (or extended Type 1) involves the hemi-sacrum [12]. Depending on the size and extent of the lesion, most hemipelvectomies include resection of more than one of these zones [13]. Advances in chemotherapy and surgical techniques have allowed for safer IH to be performed, and, additionally, they have improved the overall survival of patients after IH [10,14,15].

Post-operative pain management in this patient population can be challenging; however, it is vital in order to obtain maximal functional outcome. Previous studies have described post-operative pain and analgesia after hemipelvectomy in adult patients [1,16,17], and phantom limb pain in children undergoing amputation [18,19,20]; however, there is a paucity of data on the evolution of pain and analgesia use pertaining to both internal and external hemipelvectomies in children. We aim to describe the incidence, severity, and evolution of perioperative pain in pediatric oncology patients undergoing hemipelvectomy, and, additionally, we aim to detail the analgesic regimens used for these patients in the perioperative course.

## 2. Materials and Methods

This study was approved by the University of Texas, M.D. Anderson Cancer Center (MDACC) Institutional Review Board. We conducted a retrospective chart review of cancer patients, aged 21 years and under, who underwent hemipelvectomy at MDACC from 2018 to 2021. Patients were identified from a database maintained by the orthopedic oncology department, and they were screened to include only ages less than 21 years. We extracted demographic and clinical data from the electronic medical record (EMR).

Primary outcome measures included the evolution of pain throughout the perioperative course, measured on post-operative days 0–7, day 30, and day of discharge, and the route, type, dose, and duration of analgesia regimens utilized during these same time points. Pain was assessed via Numeric Pain Rating Scale [21]. Analgesic routes were categorized as enteral, parenteral, or regional, and classified by drug type. Opioids were measured in oral (PO) morphine equivalent daily dose in milligrams (mg) (MEDD) per kilogram (kg) on each of the aforementioned post-operative days [22,23].

Secondary outcome measures included the measurement of perioperative functional status, assessed at the same time points by the Activity Measurement for Postacute Care (AM-PAC) score [24,25], and length of hospitalization, length of intensive care unit (ICU) stay, and survival. The AM-PAC score is a 41-item clinical measurement tool that assesses patients’ execution of discrete daily tasks in their environments across major content domains defined by the WHO International Classification of Functioning, Disability, and Health [24,26]. This provides comprehensive measurement of functional activity in three broad categories: Applied Cognition, Personal Care and Instrumental, and Physical and Movement. AM-PAC scores were completed by physical therapy (PT) and occupational therapy (OT) specialists.

Additional demographic and perioperative data collected included cancer diagnosis, age, sex, type of hemipelvectomy, surgical margins, necrosis percentage, adjuvant chemotherapy, perioperative labs, blood product transfusions, post-operative complications, PT, OT, nutrition and psychology consults, time to first ambulation, time to full diet, time to first bowel movement, time to tubes and drains removal, and 30-day readmission status.

Patient demographics and clinical characteristics were described and analyzed using means and medians for continuous variables, and percentages for categorical variables. Baseline variables were summarized as means with standard deviations or as frequencies with percentages.

## 3. Results

### 3.1. Patient Demographics

Patient demographics are summarized in Table 1. During the three-year study period, eight patients with cancer, who were between ages seven and fourteen years, underwent hemipelvectomy. Patient characteristics are summarized in Table 1. The mean age of the patient population was 13 ± 2.93 years. The mean weight was 46.18 ± 2.68 kg. Four (50%) of the patients were female. Six patients presented with Osteosarcoma, and two patients presented with Ewings sarcoma. One of the patients with Osteosarcoma displayed a history of completely resected lung metastases, while all of the others had localized disease.

### 3.2. Hemipelvectomy Characteristics

Hemipelvectomy characteristics are summarized in Table 2. Four patients (50%) received internal hemipelvectomy (IH), and four patients (50%) received external hemipelvectomy (EH). The most common zone of hemipelvectomy was Type 1/2, with five cases (62.5%). Four (50%) of the hemipelvectomies were left-sided. Three (37.5%) of the patients received reconstruction. All surgical margins were noted to be free of disease. All patients had received neoadjuvant chemotherapy as directed by their histologic diagnosis. The mean time to surgical drain removal was 23.41 ± 11.12 days. The mean time to foley catheter removal was 12.38 ± 6.49 days. The mean length of ICU stay was 12.25 ± 8.49 days. Mean length of hospitalization was 29.39 ± 14.17 days. Complications temporally related to hemipelvectomy occurred in four patients (50%), and these included a deep venous thrombosis requiring thrombectomy, reoperation for flap compromise/necrosis, and superficial wound infection requiring irrigation and debridement. All patients survived the post-operative period. The 12-month event-free survival (EFS) and overall survival (OS) were 87.5% (*n* = 7) and 100%, respectively; the 24-month EFS and OS were 75% (*n* = 6) and 87.5% (*n* = 7), respectively; and the 36-month EFS and OS were 62.5% (*n* = 5) and 75% (*n* = 6), respectively. The cause of death in the two deceased patients were related to metastatic disease progression.

### 3.3. Pain and Analgesia Data

Pain and analgesia data are summarized in Table 3. On average across all patients, pain scores down trended throughout the post-operative period. The mean pain score was highest on post-operative day (POD) 0 at 3 ± 3.75, POD5 at 3 ± 2.33, and POD 30 at 3 ± 3.02, and it was the lowest at 0.75 ± 1.09 on day of discharge. This trend is displayed in Figure 1. On average across all patients, the mean pain score throughout the time points was 2.16 ± 1.09. Regarding patients who underwent IH, the mean pain score throughout the time points was 1.62 ± 0.85, and, regarding patients who underwent EH, the mean pain score throughout the time points was 1.77 ± 0.96. All patients were transitioned from parenteral to enteral analgesics by day of discharge. Post-operative regional anesthesia was used in five patients (62.5%); four patients received hydromorphone with bupivacaine (50%), one patient received lidocaine (12.5%), and one patient received an unspecified amide (12.5%). All patients received acetaminophen and opioids. Five patients (62.5%) received ketamine intraoperatively, and one of these patients (12.5%) received ketamine post-operatively. Five patients (62.5%) received gabapentin post-operatively. One patient (12.5%) received non-steroidal antiflammatory drugs—ketorolac and celecoxib. The most common agents used for opioid analgesia were fentanyl (eight subjects, 100%), methadone (six subjects, 75%), and hydromorphone (five subjects, 62.5%). The mean PO MEDD in mg/kg noted as “MEDD/kg” was 2.64 ± 1.43. On average across all post-operative days, the mean MEDD/kg was highest on POD5, at 5.03 ± 6.06, and it was the lowest post-operatively at 0.75 ± 0.74 on day of discharge. This trend is displayed in Figure 2. On average across all post-operative days, the mean MEDD/kg for patients who underwent IH was 1.82 ± 1.58, and the mean MEDD/kg for patients who underwent EH was 3.46 ± 3.61 (*p* = 0.5). The mean duration on analgesics was 31.63 ± 13.47 days. All patients were converted to oral opioids by day of discharge.

### 3.4. Rehabilitation Data

Rehabilitation data are summarized in Table 4. On average across all patients, AM-PAC scores increased throughout the post-operative period. This trend is displayed in Figure 3. AM-PAC scores were not assessed in most patients until POD2. The mean AM-PAC score was 12.42 ± 1.31. AM-PAC score was highest on day of discharge, at 18.5 ± 2.45. Six (75%) of the patients were ambulating after hemipelvectomy. For these patients, the mean time to ambulation was 5.33 ± 2.94 days. For the two patients who were non-ambulatory, one patient was non-ambulatory pre-operatively due to a severe pain, and post-operative rehabilitation was complicated by severe autism; the other patient had tumor involvement of the lumbosacral spine, resulting in a neurologic deficit. Four patients (50%) were discharged ambulating without assistance. For these patients, the mean time to ambulation without assistance was 24.75 ± 9.29 days. The mean time to full diet was 4.75 ± 3.45 days. The mean time to first bowel movement was 7.88 ± 3.64 days. PT and OT were consulted in all patients. The mean time to PT consult visit was 1 ± 1.21 days. The mean time to OT consult visit was 1.13 ± 1.21 days. All patients were evaluated by PT and OT daily. Nutrition was consulted in seven patients (87.5%). All nutrition consults were within one day. The mean frequency of nutrition evaluation was every 6.71 ± 3.08 days. Pre-operative psychology consult occurred in three patients (37.5%), while post-operative consult occurred in an additional two patients (25%). The mean time to psychology post op visit was 4 ± 7.07 days. The mean frequency of psychology evaluation was every 12.5 ± 3.54 days.

### 3.5. Hematologic and Blood Product Data

Hematologic and blood product data are summarized in Table 5. Regarding laboratory values, the mean pre-operative white blood cell count was 6.06 ± 3.00 leukocytes/mL. The mean pre-operative hemoglobin level was 10.24 ± 1.36 mg/dL. The mean pre-operative platelet count was 242.75 ± 64.31 K/µL. The mean pre-operative international normalized ratio (INR) was 1.04 ± 0.12. The mean post-operative white blood cell count was 11.74 ± 6.27 leukocytes/mL. The mean post-operative hemoglobin level was 10.36 ± 1.55 mg/dL. The mean post-operative platelet count was 110.75 ± 34.39 K/µL. The mean post-operative INR was 1.21 ± 0.16. The mean post-operative fibrinogen was 298 ± 174.38. All patients received blood product transfusions. The most common type of blood product transfusions received perioperatively were packed red blood cells (8 subjects, 100%), fresh frozen plasma (8 subjects, 100%), and platelets (4 subjects, 50%). The mean number of blood product transfusions each patient received was 23.5 ± 16.03.

## 4. Discussion

This retrospective study assessed the evolution of pain, analgesia regimens, and functional status of pediatric cancer patients after hemipelvectomy. Our findings describe a pediatric oncologic cohort of hemipelvectomy patients with 100% free surgical margins, mean pain scores and opioid use that decreased throughout the post-operative course, and functional status that increased throughout the post-operative course, with 75% of patients ambulating and surviving post-operatively. The post-operative care of these patients at our institution involves collaboration amongst pediatric orthopedic oncology, pediatric oncology, pediatric anesthesia, pediatric supportive care, and pediatric critical care, as well as nursing, PT, OT, nutrition, and psychology.

Acute pain is a hallmark of pediatric pelvic sarcoma patients, occurring both at presentation and in the post-operative period following hemipelvectomy [6,19,27]. Post-operative pain in these patients involves a complex combination of both nociceptive and neuropathic components, including somatic pain related to nerve infiltration by the tumor itself, external nerve compression and bone pain, as well as post-operative inflammatory changes [6,19,27,28]. In 1986 the World Health Organization (WHO) published analgesic guidelines for the treatment of cancer pain based on a three-step ladder, including prevention of the onset of pain, simplicity of administration, and ability to be individualized [28,29,30]. However, there remains limited data regarding the most effective analgesic regimen for the cohort of patients described in this study. Interprofessional collaboration and perioperative nursing care are imperative for the reduction of acute pain in this population [20]. Analgesia in these patients, at our institution, is managed in collaboration between orthopedic surgical oncology, pediatric anesthesia, pediatric supportive care, and pediatric critical care.

The mean pain scores peaked on POD0, POD5, and POD30, while peak opioid administration peaked on POD5. Pain on POD0 is most likely due to hemipelvectomy surgical pain. It is striking to observe that both the mean pain score and opioid administration peaked on POD5, which coincides with the mean time to ambulation of 5.33 ± 2.94 days. Patients who underwent EH received a non-significant increased opioid dosage, but they displayed similar pain scores compared with IH patients across hospitalization. The small sample size may have precluded the detection of significant differences in opioid dose.

Pain management was accomplished in all patients through a combination of opioid and non-opioid analgesic agents. Acetaminophen, given its well-tolerated side effect profile in the absence of hepatic insufficiency, is a safe medication that acts centrally, crossing the blood-brain barrier in large amounts, and satisfies all components of the WHO three-step ladder in the prevention and treatment of mild post-operative pain in our cohort [29,31]. Beyond mild pain, the WHO has described opioids as the mainstay of therapy for cancer-related pain, from mild to severe. Opioids target multiple receptors both in the central nervous system (CNS) and outside of the CNS, and they are most effective at targeting nociceptive pain [29,32]. All patients received fentanyl and most received hydromorphone and/or methadone. Fentanyl is a useful analgesic in the operative and critical care settings given its rapid onset of action and short half-life [29], allowing for easy titratability and quick relief of pain in the early perioperative period. Conversely, hydromorphone and methadone have longer half-lives, allowing for less frequent monitoring and dosing, which make them excellent candidates for the later post-operative period in preparation for discharge [29]. Lastly, the addition of acetaminophen to opioids results in a synergistic effect, such that lower doses of opioids can produce pain relief with fewer side effects [29].

The majority of patients received gabapentin. While opioids target nociceptive pain, up to 90% of patients can develop peripheral neuropathy from neurotoxic chemotherapy [33]. Gabapentin is an analogue of γ–amino butyric acid (GABA) that binds to calcium channels, blocking calcium influx into nerve terminals, thereby decreasing neurotransmitter release [33,34]. It has been demonstrated to provide effective analgesia for different manifestations of neuropathic pain, including burning, allodynia, dysesthesia, and lancinating pain [34]. Gabapentin has also been shown to improve quality of life measures and improve sleep in patients with cancer-related neuropathic pain [33]. Its side effect profile includes somnolence and dizziness, is generally well-tolerated, and may resolve over time [33].

Most patients received ketamine, all of whom received ketamine intraoperatively, as well as one who remained on a ketamine infusion until POD3. Ketamine is a useful analgesic adjunct in the operative and critical care settings, functioning pharmacologically as an NMDA-receptor antagonist [35]. It has the added benefit of not causing respiratory depression or physical dependency, unlike opioids [35]. However, this medication is associated with several other unpleasant side effects, including hallucinations, anxiety, stimulation of the cardiovascular system, and vomiting. It provides a short duration of analgesia, therefore is rarely used outside of the aforementioned settings [35].

One patient received NSAIDs. Previous studies have described the role of cyclooxygenase-2 (COX-2) in bone graft healing, and thus NSAIDs, which reversibly inhibit COX-2, are usually avoided in the perioperative period following hemipelvectomy [36,37]. The one patient who received NSAIDs did not have any complications related to bone graft healing.

Apart from parenteral and enteral analgesics, most patients also received regional anesthesia. Regional anesthesia involves blockage of spinal and peripheral nerve transmission via afferent pain signaling pathways, potentially reducing post-operative pain and opioid consumption [38,39]. Local anesthetics are classically divided into two categories: esters and amides [40]. Esters local anesthetics contain an ester linkage and are metabolized by pseudocholinesterase in the plasma, whereas amide local anesthetics contain an amide linkage and are metabolized in the liver by microsomal enzymes [39]. All local anesthetics administered in our cohort belong to the amide category.

A total of 75% of patients were ambulating post-operatively, and 50% of patients were discharged ambulating without assistance. Figure 3 demonstrates that mean functional status, measured via AM-PAC scores, increased throughout the post-operative course. These results may be explained by the fact that our patient cohort benefitted from early consultation with PT and OT, with mean time to consult at 1 ± 1.21 days and 1 ± 1.21 days, respectively, while they further benefitted from daily evaluation and management from both services. Early mobility and ambulation are encouraged in the post-operative rehabilitation of pediatric hemipelvectomy patients in order to mitigate deconditioning. Considerations for the timing of ambulation trials include level of amputation, ability to bear weight, voluntary motor control, cognition, adequate analgesia with minimal sedation, wound healing, and well-controlled lower extremity edema. Bedside staff assistance and durable medical equipment, such as prosthetics, orthotics, or walkers, may help facilitate ambulation in select cases [1,41,42].

There are several strengths to this study. Most importantly, this is the first study describing the evolution of pain, analgesic regimens, and functional status of pediatric cancer patients after hemipelvectomy in the perioperative period. We described a patient cohort with mean pain scores of 3 at their peak, most of whom were ambulating, with improved functional status on average, and all on oral analgesics by discharge. Consistent with the published experience, we use discharge criteria, regarding pediatric hemipelvectomy patients, that includes adequate pain control on oral analgesics, evidence of wound healing, bowel/bladder continence, and adequate functional performance for activities of daily living either independently or with the assistance of a caregiver [1,41,43].

While retrospective cohort studies can provide insight into the ideal management of pain, there are limitations. This was a single institution study with a small number of unique patients. Given the small cohort of patients, linear regression analysis was not substantially powered to derive associations between outcomes. In addition, the retrospective data extraction did not allow for collection of patient reported outcomes regarding health related quality of life.

Further investigation at MDACC within this scope will focus on our prospective Enhanced Recovery Program [44], which aims to standardize perioperative treatment plans for pediatric surgical patients in order to optimize opioid sparing pain control, improve mobility, reduce length of stay, and improve other surgical outcomes. We also plan to compare pain control and opioid exposure, in patients following our ERP guideline, to patients in our historical cohort.

## 5. Conclusions

Acute pain is a hallmark of pediatric pelvic sarcoma patients, occurring both at presentation and in the post-operative period. The combination of perioperative acetaminophen with opioids, as well as adjunctive regional analgesia, non-steroidal anti-inflammatory drugs, gabapentin, and/or ketamine in select patients, appears to be an effective analgesic regimen for these patients. Early and frequent evaluation and treatment of pain is imperative in order to enable these patients to participate in essential physical and occupational therapy to reach their optimal goal of assistance free ambulation. In conclusion, this retrospective series suggests that hemipelvectomy for pediatric patients requiring local control for pelvic bone and soft tissue malignant neoplasms, when managed with multimodal pharmacologic analgesia, is associated with acceptable pain control and positive functional outcomes.

## Figures and Tables

**Figure 1 children-09-00237-f001:**
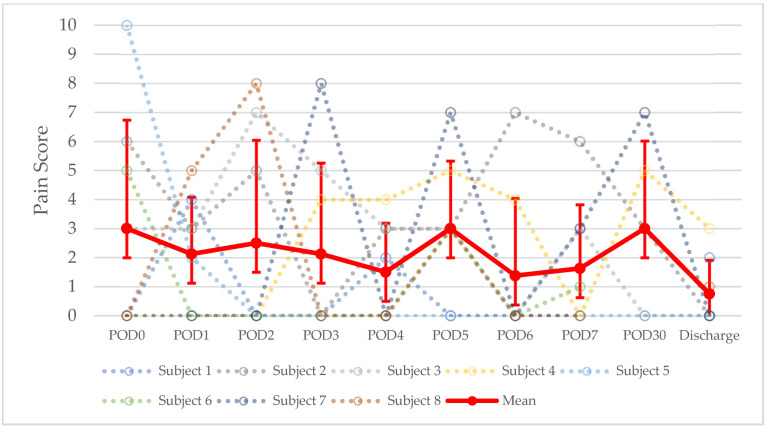
Evolution of post-operative pain scores after hemipelvectomy. POD = post-operative day.

**Figure 2 children-09-00237-f002:**
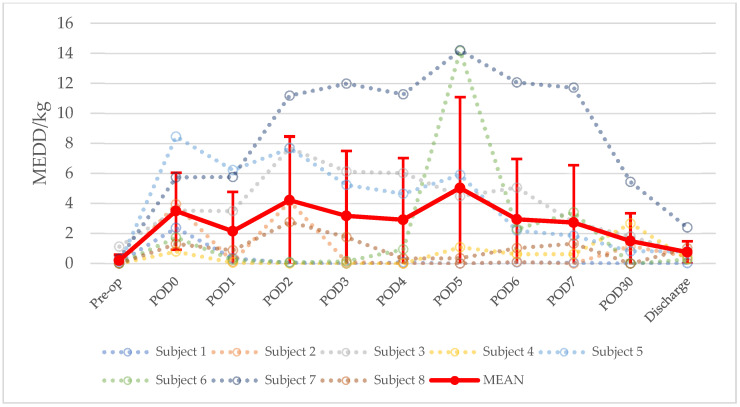
Evolution of opioid administration after hemipelvectomy. MEDD/kg = oral morphine equivalent daily dose per kilogram. Pre-op = pre-operative. POD = post-operative day.

**Figure 3 children-09-00237-f003:**
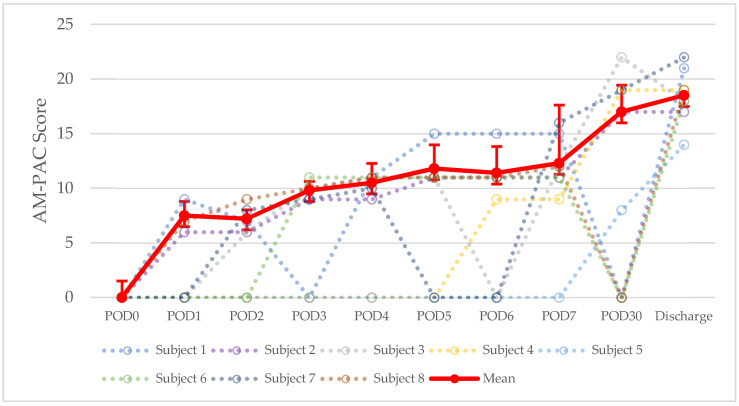
Evolution of AM-PAC scores after hemipelvectomy. POD = post-operative day.

**Table 1 children-09-00237-t001:** Patient demographics.

Characteristics	Number of Patients (%), *n* = 8
Mean patient age ± SD ^1^, years	13 ± 2.93
Mean weight ± SD, kg	46.18 ± 12.68
Female patients	4 (50)
Oncologic diagnoses	
Osteosarcoma	5 (62.5)
Osteoblastic osteosarcoma	3 (37.5)
Chondroblastic osteosarcoma	2 (25)
Ewing’s sarcoma	3 (37.5)

^1^ SD = standard deviation.

**Table 2 children-09-00237-t002:** Hemipelvectomy characteristics.

Characteristics	Number of Patients (%), *n* = 8
External hemipelvectomy	4 (50)
Internal hemipelvectomy	4 (50)
Zone	
Type 2	1 (12.5)
Type 1/2	5 (62.5)
Type 1/2/3	1 (12.5)
Type 2/3	1 (12.5)
Left-sided	4 (50)
Reconstruction	3 (37.5)
Free surgical margins	8 (100)
Mean necrosis percentage ± SD ^1^, %	76.75 ± 24.40
Received pre-operative chemotherapy	8 (100)
Doxorubicin	8 (100)
Ifosfamide	5 (62.5)
Cisplatin	5 (62.5)
Methotrexate	5 (62.5)
Received post-operative chemotherapy	7 (87.5)
Doxorubicin	4 (50)
Ifosfamide	4 (50)
Etoposide	4 (50)
Mean time to surgical drain removal ± SD, days	23.41 ± 11.12
Mean time to foley catheter removal ± SD, days	12.38 ± 6.49
Mean length of hospitalization ± SD, days	29.39 ± 14.17
Mean length of ICU stay ± SD, days	12.25 ± 8.49
Complications	4 (50)
36-month Overall Survivors	6 (75)

^1^ SD = standard deviation.

**Table 3 children-09-00237-t003:** Pain and analgesic data.

Characteristics	Number of Patients (%), *n* = 8
Mean pain score ± SD ^1^	2.16 ± 1.09
Post-operative day 0	3 ± 3.75
Post-operative day 5	3 ± 2.33
Post-operative day 30	3 ± 3.02
Enteral analgesia	8 (100)
Parenteral analgesia	8 (100)
Regional anesthesia	5 (62.5)
Hydromorphone with bupivacaine	4 (50)
Lidocaine	1 (12.5)
Unspecified amide	1 (12.5)
Ketamine	5 (62.5)
Non-steroidal anti-inflammatory drugs	1 (12.5)
Acetaminophen	8 (100)
Gabapentin	5 (62.5)
Opioids	8 (100)
Fentanyl	8 (100)
Hydromorphone	5 (62.5)
Methadone	6 (75)
Mean morphine equivalent daily dose/kg ± SD	2.64 ± 1.43
Mean duration on analgesics ± SD, days	31.63 ± 13.47

^1^ SD = standard deviation.

**Table 4 children-09-00237-t004:** Rehabilitation data.

Characteristics	Mean ± SD ^1^
AM-PAC ^2^ score	12.42 ± 1.31
Patients ambulating post-operatively, *n* (%)	6 (75)
Time to ambulation, days	5.33 ± 2.94
Patients discharged ambulating without assistance, *n* (%)	4 (50)
Time to ambulation without assistance, days	24.75 ± 9.29
Time to full diet, days	4.75 ± 3.4
Time to first bowel movement, days	7.88 ± 3.64
PT ^3^ consult, *n* (%)	8 (100)
Time to PT consult, days	1 ± 1.21
Frequency of PT evaluation, every *n* days	1 ± 0
OT ^4^ consult, *n* (%)	8 (100)
Time to OT consult, days	1.13 ± 1.21
Frequency of OT evaluation, every n days	1 ± 0
Nutrition consult, *n* (%)	7 (87.5)
Time to nutrition consult, days	1 ± 0
Frequency of nutrition evaluation, every n days	6.71 ± 3.08
Psychology consult, *n* (%)	5 (62.5)
Time to psychology post op visit, days	4 ± 7.07
Frequency of nutrition evaluation, days	12.5 ± 3.54

^1^ SD = standard deviation. ^2^ AM-PAC = activity measurement for postacute care. ^3^ PT = physical therapy. ^4^ OT = occupational therapy.

**Table 5 children-09-00237-t005:** Hematologic and blood product data.

Characteristics	Mean ± SD ^1^
Pre-operative WBC ^2^ count, leukocytes/mL	6.06 ± 3.00
Pre-operative hemoglobin, mg/dL	10.24 ± 1.36
Pre-operative platelet, K/µL	242.75 ± 64.31
Pre-operative international normalized ratio	1.04 ± 0.12
Post-operative WBC count, leukocytes/mL	11.74 ± 6.27
Post-operative hemoglobin, mg/dL	10.36 ± 1.55
Post-operative platelet count, K/µL	110.75 ± 34.39
Post-operative international normalized ratio	1.21 ± 0.16
Blood product transfusions, *n* (%)	8 (100)
Packed red blood cells, *n* (%)	8 (100)
Fresh frozen plasma, *n* (%)	8 (100)
Platelets, *n* (%)	4 (50)
Number of transfusions per patient	23.5 ± 03

^1^ SD = standard deviation. ^2^ WBC = white blood cell.

## Data Availability

The data presented in this study are available on request from the corresponding author. The data are not publicly available in order to protect the privacy of the subjects in this study.

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
