# Peer review of "Pain and Analgesia in Children with Cancer after Hemipelvectomy: A Retrospective Analysis"

_children, 2022, doi:10.3390/children9020237_

Round 1
Reviewer 1 Report
It would be good to design a further investigation to compare different sets of analgesic drugs
Author Response
- We appreciate the suggestion for comparative trials of analgesic regimens. We decided to prioritize standardizing a pain management practice guideline for our post-operative patients that minimizes IV opioids without an adverse effect on pain control in our prospective Enhanced Recovery Program (ERP).
- We added an additional sentence in the Discussion section that states:
- We also plan to compare pain control and opioid exposure in patients following our ERP guideline to our historical cohort [44].
44. Wells, S. J., Austin, M., Gottumukkala, V., Kruse, B., Mayon, L., Kapoor, R., ... & Swartz, M. C. (2021). Development of an Enhanced Recovery Program in Pediatric, Adolescent, and Young Adult Surgical Oncology Patients. Children, 8(12), 1154.
Reviewer 2 Report
- In the parts of result and discussion, discharge day may influence the type of pain control, but no description for patient discharge criteria.
- No detail about how to decide the timing of patient ambulation.
- Internal or external hemipelvectomy may cause different grade of tissue/bone damage. The pain control effect between two procedures can be explained separated.
Author Response
1. In the parts of result and discussion, discharge day may influence the type of pain control, but no description for patient discharge criteria.
- Thank you for your thoughtful review and comments. We added an additional sentence in the Discussion section that states:
- Consistent with the published experience, we use discharge criteria for pediatric hemipelvectomy patients that includes adequate pain control on oral analgesics, evidence of wound healing, bowel/bladder continence, and adequate functional performance for activities of daily living independently or with the assistance of a caregiver [1,41,43].
1. Guo Y, Fu J, Palmer JL, Hanohano J, Cote C, Bruera E. Comparison of post-operative rehabilitation in cancer patients undergoing internal and external hemipelvectomy. Arch Phys Med Rehabil. 2011 Apr;92(4):620-5. doi: 10.1016/j.apmr.2010.11.027. PMID: 21440708.
41. Robinson KM, Lackman RD, Donthineni-Rao R. Rehabilitation after hemipelvectomy. University of Pennsylvania Orthopaedic Journal 14: 61-69, 2001.
43. Schaal Wilson RE. (2015) Rehabilitation considerations for a patient with external hemipelvectomy and hemisacrectomy for recurrent soft tissue pelvic sarcoma: a case report, Physiotherapy Theory and Practice, 31:6, 433-441
2. No detail about how to decide the timing of patient ambulation.
- Thank you for your thoughtful review and comments. We added additional verbiage in the Discussion section that states:
- Early mobility and ambulation are encouraged in the post-operative rehabilitation of pediatric hemipelvectomy patients to mitigate deconditioning. Considerations for the timing of ambulation trials include level of amputation, ability to bear weight, voluntary motor control, cognition, adequate analgesia with minimal sedation, wound healing, and well-controlled lower extremity edema. Bedside staff assistance and durable medical equipment such as prosthetics, orthotics, or walkers may help facilitate ambulation in select cases [1,41,42].
1. Guo Y, Fu J, Palmer JL, Hanohano J, Cote C, Bruera E. Comparison of post-operative rehabilitation in cancer patients undergoing internal and external hemipelvectomy. Arch Phys Med Rehabil. 2011 Apr;92(4):620-5. doi: 10.1016/j.apmr.2010.11.027. PMID: 21440708.
41. Robinson KM, Lackman RD, Donthineni-Rao R. Rehabilitation after hemipelvectomy. University of Pennsylvania Orthopaedic Journal 14: 61-69, 2001.
42. Houdek MT, Kralovec ME, Andrews KL. Hemipelvectomy: high-level amputation surgery and prosthetic rehabilitation. Am J Phys Med Rehabil. 2014 Jul;93(7):600-8. doi: 10.1097/PHM.0000000000000068. PMID: 24508940.
3. Internal or external hemipelvectomy may cause different grade of tissue/bone damage. The pain control effect between two procedures can be explained separated.
- Thank you for your thoughtful review and comments. We added additional verbiage in the Results section that states:
- For patients who underwent IH, the mean pain score throughout the time points was 1.62 ± 0.85, and for patients who underwent EH, the mean pain score throughout the time points was 1.77 ± 0.96. On average across all post-operative days, the mean MEDD/kg for patients who underwent IH was 1.82 ± 1.58, and the mean MEDD/kg for patients who underwent EH was 3.46 ± 3.61.(p=0.5)
- We also added additional verbiage in the Discussion section that states:
- Patients who underwent EH received a non-significant increased opioid dosage, but similar pain scores compared with IH patients across hospitalization. The small sample size may have precluded detecting significant differences in opioid dose.
Round 2
Reviewer 2 Report
I think that the revisions of current study are enough and well done.